# A Lightweight Method to Generate Unanswerable Questions in English

**Vagrant Gautam**  **Miaoran Zhang**  **Dietrich Klakow**

Saarland Informatics Campus, Saarland University

vgautam@lsv.uni-saarland.de

## Abstract

If a question cannot be answered with the available information, robust systems for question answering (QA) should know *not* to answer. One way to build QA models that do this is with additional training data comprised of unanswerable questions, created either by employing annotators or through automated methods for unanswerable question generation. To show that the model complexity of existing automated approaches is not justified, we examine a simpler data augmentation method for unanswerable question generation in English: performing antonym and entity swaps on answerable questions. Compared to the prior state-of-the-art, data generated with our training-free and lightweight strategy results in better models (+1.6 F1 points on SQuAD 2.0 data with BERT-large), and has higher human-judged relatedness and readability. We quantify the raw benefits of our approach compared to no augmentation across multiple encoder models, using different amounts of generated data, and also on TydiQA-MinSpan data (+9.3 F1 points with BERT-large). Our results establish swaps as a simple but strong baseline for future work.

## 1 Introduction

Question answering datasets in NLP tend to focus on answerable questions (Joshi et al., 2017; Fisch et al., 2019), but unanswerable questions matter too because: (1) *real-world queries are unanswerable surprisingly often* – e.g., 37% of fact-seeking user questions to Google are unanswerable based on the Wikipedia page in the top 5 search results (Kwiatkowski et al., 2019); and (2) *identifying unanswerable questions is an essential feature of reading comprehension* – but conventional extractive QA systems typically guess at plausible answers even in these cases (Rajpurkar et al., 2018).

To aid in building robust QA systems, more datasets have begun to include unanswerable questions, e.g., the SQuAD 2.0 dataset in English (Rajpurkar et al., 2018) and the multilingual TydiQA

Figure 1: A context paragraph and an answerable seed question, which can be used to generate unanswerable questions. Examples are shown from human annotators as well as 4 automatic methods with their estimated number of training parameters. 🌟 indicates our methods.

dataset (Clark et al., 2020), both of which contain human-written answerable and unanswerable examples of extractive question answering. As human annotation is slow and costly, various models have been proposed to automate unanswerable question generation using answerable seed questions; most recently, Zhu et al. (2019) proposed training on a pseudo-parallel corpus of answerable and unanswerable questions, and Liu et al.'s (2020) state-of-the-art model used constrained paraphrasing.

Although model-generated unanswerable questions give sizeable improvements on the SQuAD 2.0 development set, Figure 1 shows that many differ from their answerable counterparts only superficially. An estimated 40% of human-written unanswerable questions also involve minor changes to answerable ones, e.g., swapping words to antonyms or swapping entities (Rajpurkar et al., 2018).

Motivated by these observations, we present a lightweight method for unanswerable question generation: performing antonym and entity swaps on answerable questions. We evaluate it with:

1. *4 metrics*: development set performance (EM and F1), as well as human-judged unanswerability, relatedness, and readability;
2. *2 datasets*: SQuAD 2.0 (Rajpurkar et al., 2018) and TydiQA (Clark et al., 2020);
3. *2 baselines*: UNANSQ (Zhu et al., 2019) and CRQDA (Liu et al., 2020); and
4. *6 encoder models*: base and large variants of BERT (Devlin et al., 2019), RoBERTa (Liu et al., 2019), and ALBERT (Lan et al., 2020).

Swapping significantly outperforms larger and more complex unanswerable question generation models on *all* metrics. Across models and datasets, our method vastly improves performance over a no-augmentation baseline. These results show that our method has potential for practical applicability and that it is a hard-to-beat baseline for future work.[1]

## 2  Related work

Unanswerability is not new to QA research, with a rich body of work typically proposing data augmentation methods or training paradigm innovations.

Papers focusing on data augmentation either generate data for adversarial evaluation (Jia and Liang, 2017; Wang and Bansal, 2018) or for training. Most work on training data generation for QA is limited to generating answerable questions, e.g., Alberti et al. (2019) and Bartolo et al. (2020, 2021), but some generate both answerable and unanswerable questions (Liu et al., 2020) or, like us, just unanswerable questions (Clark and Gardner, 2018; Zhu et al., 2019). Unanswerable questions have been shown to be particularly hard for contemporary QA models when they contain false presuppositions (Kim et al., 2023), when they are fluent and related (Zhu et al., 2019), when the context contains a candidate answer of the expected type (e.g., a date for a "When" question; Weissenborn et al., 2017; Sulem et al., 2021), and in datasets beyond SQuAD (Sulem et al., 2021). Our method is challenging for models because it generates questions that are fluent, related and unanswerable.

Different training paradigms have been proposed to more effectively use training data, e.g., adversarial training (Yang et al., 2019) and contrastive learning (Ji et al., 2022), or to tackle unanswerability and answer extraction separately, by using verifier modules or calibrators (Tan et al., 2018; Hu et al., 2019; Kamath et al., 2020). We use a conventional fine-tuning paradigm and leave it to future work to boost performance further by using our high-quality data in other paradigms.

## 3  Our augmentation methods

Inspired by the crowdworker-written unanswerable questions in Rajpurkar et al. (2018), we generate 2 types of unanswerable questions by modifying answerable ones with antonym and entity swaps. Our generated data is then filtered based on empirical results presented in Appendix B.1. We examine results for each augmentation method separately, but we also experimented with combining them in Appendix B.2. Examples of output from our methods are shown in Figure 1 and in Appendix D.

We use spaCy (Honnibal et al., 2020) for part-of-speech tagging and dependency parsing, and AllenNLP's (Gardner et al., 2018) implementation of Peters et al.'s (2017) model for named entity recognition. Using NLTK (Bird et al., 2009), we access WordNet (Fellbaum, 1998) for antonyms and lemmatization.

### 3.1  Antonym augmentation

We antonym-augment an answerable question by replacing one noun, adjective or verb at a time with its antonym. We replace a word when it exactly matches its lemma,[2] with no sense disambiguation. When multiple antonym-augmented versions are generated, we pick the one with the lowest GPT-2 perplexity (Radford et al., 2019). Thus, when we augment the question "*When did Beyonce start becoming popular?*" we choose "*When did Beyonce start becoming unpopular?*" instead of the clunky "*When did Beyonce end becoming popular?*".

To avoid creating answerable antonym-augmented questions, we do not augment adjectives in a dependency relation with a question word (e.g., "*How big are ostrich eggs?*"), and we also skip polar questions (e.g., "*Is botany a narrow science?*") and alternative questions (e.g., "*Does communication with peers increase or decrease during adolescence?*"), both of which tend to begin with an *AUX* part-of-speech tag.

---

[1]Our data and code are available at `https://github.com/uds-lsv/unanswerable-question-generation`.

[2]English's lack of rich morphology lets us avoid inflection models with little impact on how much data we can generate.

## 3.2 Entity augmentation

We entity-augment an answerable question by replacing one entity at a time with a random entity from the context document that has the same type and does not appear in the question: "*How old was Beyoncé when she met LaTavia Roberson?*" can be augmented to "*How old was Beyoncé when she met Kelly Rowland?*" but it can never be augmented to "*How old was Beyoncé when she met Beyoncé?*" When we generate multiple entity-augmented versions of a question, we randomly select one.

Intuitively, picking an entity of the same type keeps readability high as person entities appear in different contexts (*married*, *died*) than, e.g., geopolitical entities (*filmed in*, *state of*). Using entities from the same context ensures high relevance, and leaving everything else unmodified maintains the entity type of the expected answer.

## 4 Experimental setup

**Task.** We evaluate on the downstream task of *extractive question answering*, i.e., we judge an unanswerable question generation method to be better if training with its data improves an extractive QA system's performance compared to other methods. Given a question and some context (a sentence, paragraph or article), the task is to predict the correct answer text span in the passage, or no span if the question is unanswerable. Performance is measured with exact match (EM) and F1, computed on the answer span strings.

**Datasets.** We use SQuAD 2.0 (Rajpurkar et al., 2018) and the English portion of the TydiQA dataset (Clark et al., 2020) that corresponds to minimal-span extractive question answering. SQuAD 2.0 uses paragraphs as context whereas TydiQA uses full articles. To keep the TydiQA setting similar to SQuAD 2.0, we modify the task slightly, discarding yes/no questions and questions for which there is a paragraph answer but not a minimal span answer. For both datasets, we train on original or augmented versions of the training set and report performance on the development set. All data statistics are shown in Table 1.

**Models.** We experiment with base and large variants of BERT (Devlin et al., 2019), RoBERTa (Liu et al., 2019), and ALBERT (Lan et al., 2020), all trained with HuggingFace Transformers (Wolf et al., 2020); see Appendix A for further details.

| Data | Answerable | Unanswerable |
|---|---|---|
| SQuAD 2.0 | | |
| Training data | 86,821 | 43,498 |
| + UNANSQ | + 0 | + 69,090 |
| + CRQDA | + 0 | + 124,085 |
| + Antonym (ours) | + 0 | + 34,180 |
| + Entity (ours) | + 0 | + 47,624 |
| Development data | 5,928 | 5,945 |
| TydiQA-MinSpan (English) | | |
| Training data | 3,696 | 4,953 |
| + Antonym (ours) | + 0 | + 880 |
| + Entity (ours) | + 0 | + 2,808 |
| Development data | 495 | 477 |

Table 1: Number of answerable and unanswerable questions with the SQuAD 2.0 and TydiQA-MinSpan datasets and the available augmentation methods.

## 5 Comparison with previous SQuAD 2.0 augmentation methods

We compare methods on their $BERT_{large}$ performance using 2 strong baselines for unanswerable question generation: UNANSQ (Zhu et al., 2019) and the state-of-the-art method CRQDA (Liu et al., 2020). We use publicly-released unanswerable questions for both methods, which only exist for SQuAD 2.0. In theory, CRQDA can generate both answerable and unanswerable questions but we only use the latter for an even comparison and because only these are made available.

### 5.1 Main result

| Training Data | EM (↑) | F1 (↑) |
|---|---|---|
| Baseline (no aug.) | $78.0_{\pm 0.3}$ | $81.2_{\pm 0.4}$ |
| + UNANSQ | $77.8_{\pm 0.6}$ | $81.0_{\pm 0.5}$ |
| + CRQDA | $79.1_{\pm 0.4}$ | $82.0_{\pm 0.4}$ |
| + Antonym (ours) | $79.3_{\pm 0.2}$ | $82.4_{\pm 0.3}$ |
| + Entity (ours) | $80.7_{\pm 0.1}$ | $83.6_{\pm 0.0}$ |

Table 2: SQuAD 2.0 development set results (EM/F1) with different data augmentation methods, averaged over 3 random seeds when fine-tuning $BERT_{large}$. Coloured cells indicate significant improvements over CRQDA according to a Welch's t-test ($\alpha = 0.05$).

As the results in Table 2 show, our proposed data augmentation methods **perform better than other more compute-intensive unanswerable question generation methods**. Entity augmentation is more effective than antonym augmentation; anecdotally, some samples of the latter are semantically incoherent, which models might more easily identify.

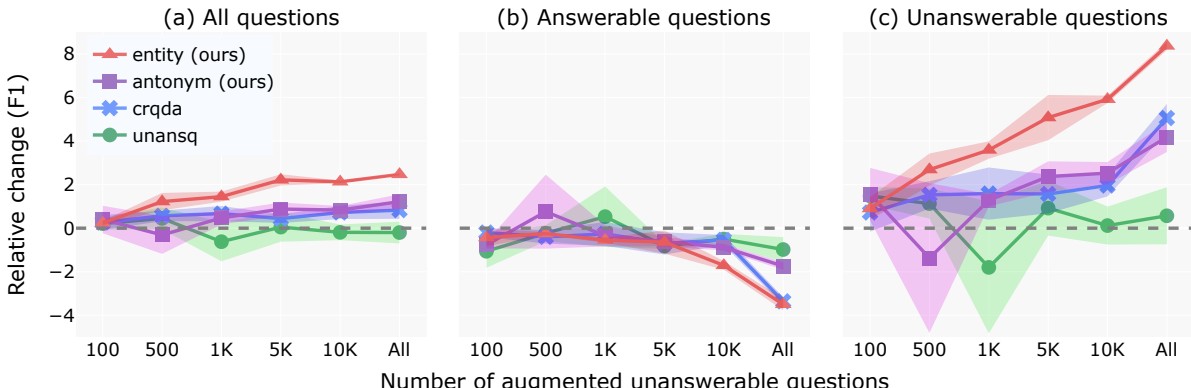

Figure 2: Relative change in BERT$_{large}$'s F1 score on all, answerable and unanswerable questions in the SQuAD 2.0 development set, varying the amount of data generated with UNANSQ, CRQDA, antonym and entity augmentation.

| Method | Unanswerability (↑) | Relatedness (↑) | Readability (↑) |
| --- | --- | --- | --- |
| (Range) | $(0.0-1.0)$ | $(0.0-1.0)$ | $(1.0-3.0)$ |
| UNANSQ (Zhu et al., 2019) | 0.56 | **0.98** | 2.46 |
| CRQDA (Liu et al., 2020) | **0.91** | 0.61 | 1.40 |
| Antonym + entity (ours) | 0.78 | **0.97** | **2.69** |
| Crowdworkers (Rajpurkar et al., 2018) | 0.83 | 0.95 | 2.91 |

Table 3: Results of human evaluation of unanswerable question generation methods with 3 annotators. Inter-annotator agreement (Krippendorff's $\alpha$) is 0.67 for unanswerability, 0.60 for relatedness, and 0.74 for readability.

We find our results to be particularly compelling given that our method is training-free and implemented in less than 150 lines of Python code, compared to, for example, CRQDA, which uses a QA model and a transformer-based autoencoder with a total of 593M training parameters (Liu et al., 2020).

## 5.2 Data-balanced ablation study

As the 4 methods under comparison each generate a different number of unanswerable questions, we perform a data-balanced ablation study by training models with 100, 500, 1K, 5K and 10K randomly-selected samples from all methods.

As Figure 2(a) shows, our simpler structural methods **perform comparably with or better than more complex methods, even with less data**.

When split by answerability, all methods show some degradation on answerable questions in the development set, as shown in Figure 2(b); like Ji et al. (2022), we find that focusing on unanswerable question generation leads to a **tradeoff between performance on unanswerable and answerable questions**. We hypothesize that this tradeoff occurs as a result of overfitting to unanswerable questions as well as data noise, i.e., generated questions that are labelled unanswerable but are actually answer-

able, which might lead the model to abstain more on answerable questions at test time.

While our results show the effectiveness of augmented unanswerable questions at improving unanswerability, they also highlight the need to ensure that this boost does not come at the cost of question *answering*. Using less augmented data might help with this; 5K entity-augmented samples vastly improve unanswerable question performance at little cost to answerable ones.

## 5.3 Human evaluation

We perform an additional human evaluation of the unanswerable question generation methods using the following 3 criteria, based on Zhu et al. (2019):

1. *Unanswerability* $(0.0-1.0)$: 0 if the generated question is answerable based on the context, 1 if it is unanswerable;
2. *Relatedness* $(0.0-1.0)$: 0 if the question is unrelated to the context, 1 if it is related;[3]
3. *Readability* $(1.0-3.0)$: 1 if the question is incomprehensible, 2 for minor errors that do not obscure meaning, 3 for fluent questions.

---

[3]Zhu et al. (2019) evaluate relatedness on a 1-3 scale, comparing each question to a context paragraph and an input question. We use a binary scale as we do not have paired input questions for CRQDA and human-written questions.

| Dataset | Model | Baseline (no aug.) | | Antonym | | Entity | |
|---|---|---|---|---|---|---|---|
| | | EM (↑) | F1 (↑) | EM (↑) | F1 (↑) | EM (↑) | F1 (↑) |
| SQuAD 2.0 | $BERT_{base}$ | $72.7_{\pm0.3}$ | $76.0_{\pm0.3}$ | $73.9_{\pm0.7}$ | $77.0_{\pm0.9}$ | $76.0_{\pm0.4}$ | $79.0_{\pm0.5}$ |
| | $BERT_{large}$ | $78.0_{\pm0.3}$ | $81.2_{\pm0.4}$ | $79.3_{\pm0.2}$ | $82.4_{\pm0.3}$ | $80.7_{\pm0.1}$ | $83.6_{\pm0.0}$ |
| | $RoBERTa_{base}$ | $78.7_{\pm0.1}$ | $81.8_{\pm0.1}$ | $79.2_{\pm0.1}$ | $82.2_{\pm0.1}$ | $79.7_{\pm0.2}$ | $82.6_{\pm0.1}$ |
| | $RoBERTa_{large}$ | $85.8_{\pm0.2}$ | $88.8_{\pm0.2}$ | $85.9_{\pm0.2}$ | $88.9_{\pm0.2}$ | $85.7_{\pm0.1}$ | $88.6_{\pm0.1}$ |
| | $ALBERT_{base}$ | $79.3_{\pm0.1}$ | $82.4_{\pm0.1}$ | $79.3_{\pm0.2}$ | $82.3_{\pm0.1}$ | $80.0_{\pm0.2}$ | $82.9_{\pm0.1}$ |
| | $ALBERT_{large}$ | $82.1_{\pm0.2}$ | $85.2_{\pm0.1}$ | $82.2_{\pm0.2}$ | $85.2_{\pm0.2}$ | $82.3_{\pm0.2}$ | $85.1_{\pm0.1}$ |
| TydiQA-MinSpan (English) | $BERT_{base}$ | $48.5_{\pm0.5}$ | $51.6_{\pm0.7}$ | $58.7_{\pm0.7}$ | $61.4_{\pm0.7}$ | $58.9_{\pm1.2}$ | $61.2_{\pm1.4}$ |
| | $BERT_{large}$ | $51.4_{\pm0.8}$ | $54.4_{\pm0.7}$ | $61.2_{\pm0.3}$ | $63.7_{\pm0.4}$ | $60.7_{\pm1.4}$ | $62.6_{\pm1.7}$ |

Table 4: Our methods give statistically significant improvements (coloured cells) across multiple encoder models on both SQuAD 2.0 and TydiQA-MinSpan data, compared to no augmentation. EM and F1 results are averaged over 3 random seeds and significance is measured using a Welch's t-test with $\alpha = 0.05$.

100 context paragraphs are sampled from SQuAD 2.0 along with 4 questions per paragraph – 1 crowdworker-written question from the original dataset, and 1 question from each of the following automated methods: UNANSQ, CRQDA, and our method (a combination of antonym- and entity-augmented questions). This gives a total of 400 questions, evaluated by 3 annotators. Complete annotator instructions are provided in Appendix C.

The evaluation results (Table 3) show our method to be an all-rounder with **high relatedness, near-human unanswerability, and the highest readability of any automatic method**. UNANSQ-generated questions are related and readable but a whopping 44% of them are answerable, while CRQDA only shines at unanswerability by generating unrelated gibberish instead of well-formed questions (52% less readable and 36% less related than crowdworker-written questions, compared to ours – 5% less readable but 2% *more* related). Despite their higher unanswerability, the CRQDA questions are not as beneficial to training, suggesting a compositional effect: unanswerability, relatedness and readability *all* play a role together and it is important for generation methods to do reasonably well at all of them.

## 6 Beyond SQuAD 2.0 and BERT-large

To more robustly evaluate our augmentation methods, we experiment with more models of multiple sizes (ALBERT and RoBERTa) as well as with an additional dataset (TydiQA-MinSpan).

Table 4 shows that our method benefits SQuAD 2.0 performance across model types, but we note that on RoBERTa and ALBERT, our approach mainly benefits small models, as larger models already have strong baseline performance. Using BERT models, the results on TydiQA show very large improvements over the baselines, with F1 and EM improving by 8-10 points on average.

## 7 Conclusion and future work

Our lightweight augmentation method outperforms the previous state-of-the-art method for English unanswerable question generation on 4 metrics: development set performance, unanswerability, readability and relatedness. We see significant improvements in SQuAD 2.0 and TydiQA-MinSpan performance (over a no-augmentation baseline) across multiple encoder models and using different amounts of generated data. Overall, we find that when it comes to unanswerable question generation, *simpler is better*. We thus hope that future work justifies its complexity against our strong baseline.

Although we have shown that entity-based augmentation creates data that is useful for models to learn from, it is still unclear *why*. Several of our examples seem to contain false presuppositions, e.g., "When did Christopher Columbus begin splitting up the large Bronx high schools?" Kim et al. (2023) term these "questionable assumptions," and find them to be challenging for models like MACAW, GPT-3 and Flan-T5. While Sugawara et al. (2022) studies what makes answerable multiple-choice questions hard for models, we still do not know what makes *unanswerable* questions hard, and how this relates to domain, model type and size.

Beyond unanswerable question generation and even question answering, we hope our work encourages NLP researchers to consider whether simpler approaches could perform competitively on a task before using sledgehammers to crack nuts.

**Limitations**

**Heuristic unanswerability.** By generating unanswerable questions with surface-level heuristic swaps instead of deep semantic information, we sometimes end up with answerable questions. Four real examples of our method's failure modes are:

- **Conjunctions**: Given the context '*Edvard Grieg, Nikolai Rimsky-Korsakov, and Antonín Dvořák echoed traditional music of their homelands in their compositions*' and the seed question '*Edvard Grieg and Antonin Dvorak used what kind of music in their compositions?*', the entity-augmented '*Edvard Grieg and Nikolai Rimsky used what kind of music in their compositions?*' is answerable.

- **Commutative relations**: As marriage is commutative, antonym-augmenting the seed question '*Chopin's father married who?*' with the context '*Fryderyk's father, Nicolas Chopin, [...] in 1806 married Justyna Krzyżanowska*' results in the still-answerable '*Chopin's mother married who?*'

- **Information is elsewhere in the context**: With the context '*Twilight Princess was launched in North America in November 2006, and in Japan, Europe, and Australia the following month*' and the seed question, '*When was Twilight Princess launched in North America?*', entity augmentation generates the technically answerable '*When was Twilight Princess launched in Japan?*' Note that this is not answerable using *extractive* QA systems.

- **Other forms of polar questions**: We do not filter out some less common forms of polar questions, e.g., '*What beverage is consumed by more people in Kathmandu, coffee or tea?*' Here, the antonym-augmented version, '*What beverage is consumed by less people in Kathmandu, coffee or tea?*' is still answerable.

Based on our human evaluation (Table 3), we estimate the level of noise of our method at around 20%. Although we cannot provide guarantees on the unanswerability of our generated questions, our goal was to show that a lightweight method can outperform more complex methods that *also do not provide such guarantees*. Thus, we find our near-human level of noise acceptable for the task.

**Limited diversity.** As we rely on swaps, our generated augmented data is syntactically very close to the original data. We do not evaluate the diversity of our generated questions compared to human-written unanswerable questions, but similar to Rajpurkar et al. (2018), we find a qualitative gap here, and leave an exploration of this as well as its impact on performance to future work.

**Depending on existing tools.** Our methods are limited by the off-the-shelf tools we rely on. We found that POS tagging and dependency parsing were notably worse for questions compared to statements, reflecting the under-representation of questions in treebanks and part-of-speech corpora.

To ensure that entities are swapped to completely different entities, we experimented with both coreference analysis tools and substring matching (i.e., assuming that "*Beyoncé Giselle Knowles*" and "*Beyoncé*" refer to the same entity). Our substring matching heuristic is both faster and more accurate, but unfortunately both approaches struggle with diacritics and cannot identify that "*Beyoncé*" and "*Beyonce*" refer to the same person.

**Other datasets and languages.** SQuAD and TydiQA are based on Wikipedia data about people, places and organizations. This lets entity-based augmentation shine, but our methods may work less well on other domains, e.g., ACE-whQA (Sulem et al., 2021), and our conceptualization of unanswerability is specific to extractive QA.

Like many methods designed for English, ours relies on simple swaps that fail on morphologically more complex languages, c.f., Zmigrod et al. (2019). In German, for instance, we might need to re-inflect some antonyms for case, number and grammatical gender. Even entity swaps may be less straightforward, sometimes requiring different prepositions, e.g., the English sentences "*She drives to [Singapore, Switzerland, Central Park]*" would be "*Sie fährt [nach Singapur, in die Schweiz, zum Central Park]*" in German.

Furthermore, our approach for excluding questions for antonym augmentation is syntax-specific in its use of part-of-speech and dependency information. Though this approach would transfer to a syntactically similar language like German, it would not work on Hindi, where polar questions are indicated by the presence of the particle *kya:* in almost any position (Bhatt and Dayal, 2020).

## Ethics statement

Teaching models to abstain from answering unanswerable questions improves the robustness and reliability of QA systems. Working on unanswerability is thus a way of directly addressing the possible harms of QA systems giving incorrect results when being used as intended.

Additionally, our paper presents a more sustainable approach for unanswerable question generation, heeding Strubell et al.'s (2019) call to use computationally efficient hardware and algorithms.

We chose not to employ Amazon Mechanical Turk workers due to its history of exploitative labour practices (Williamson, 2016; Kummerfeld, 2021), and instead employed annotators who are contracted with the authors' institution and paid a fair wage. Our data and annotation tasks posed negligible risk of harm to the annotators.

## Acknowledgements

The authors are grateful for Eileen Bingert and AriaRay Brown's diligent annotation, to Marius Mosbach for proofreading and mentorship, and to Alexander Koller and our anonymous conference and workshop reviewers for their suggestions to improve this work. We dedicate this paper to China Restaurant Saarbrücken, where this collaboration began with a serendipitous fortune cookie that said:

*New visions create power and self-confidence.*

The authors received funding from the BMBF's (German Federal Ministry of Education and Research) SLIK project under the grant 01IS22015C, and from the DFG (German Research Foundation) under project 232722074, SFB 1102.

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

## A Implementation details

For our experiments, we initialize the following models with checkpoints from the Huggingface Transformers Library (Wolf et al., 2020): *bert-base-uncased*, *bert-large-cased*, *roberta-base*, *roberta-large*, *albert-base-v2*, and *albert-large-v2*. We use the SQuAD 2.0 hyperparameters that were suggested in the papers for each model (Devlin et al., 2019; Liu et al., 2019; Lan et al., 2020) and report them in Table 5. We train models with 3 random

| Hyperparameter | Value |
|---|---|
| Learning Rate | 5e-5 (BERT) |
| | 1.5e-5 (RoBERTa) |
| | 3e-5 (ALBERT) |
| Batch Size | 48 |
| Epochs | 2 |
| Max Seq Length | 384 |
| Doc Stride | 128 |

Table 5: Model fine-tuning hyperparameters.

seeds (42, 31, and 53). Training was conducted on a single NVIDIA A100 GPU.

We download augmented datasets from GitHub for UNANSQ[4] and CRQDA[5] and fine-tune models from scratch with the hyperparameter settings above for a fair comparison. To control for the effect of different codebases and hyperparameters, we compare the experimental results from the CRQDA codebase[6] with those of our own in Table 6, showing that our improvements are consistent.

| Training Data | EM ($\uparrow$) | F1 ($\uparrow$) |
|---|---|---|
| CRQDA codebase | | |
| Baseline (no aug.) | 78.2 | 81.4 |
| + CRQDA | 78.7 | 81.5 |
| + Entity (ours) | **80.0** | **82.9** |
| Our codebase | | |
| Baseline (no aug.) | $78.3_{\pm 0.3}$ | $81.2_{\pm 0.4}$ |
| + CRQDA | $79.1_{\pm 0.4}$ | $82.0_{\pm 0.4}$ |
| + Entity (ours) | $\mathbf{80.7}_{\pm 0.1}$ | $\mathbf{83.6}_{\pm 0.0}$ |

Table 6: Comparing codebases on their SQuAD 2.0 development set performance when fine-tuning BERT$_{large}$ on unaugmented and augmented training data.

## B Filtering and combining augmentation methods

This appendix presents our ablation experiments with filtering generated data for each augmentation strategy and combining both strategies together.

### B.1 Filtering augmented data

Table 7 shows our experiments with different filtering strategies when we generate multiple augmented versions of a single answerable question.

---

[4] https://github.com/dayihengliu/CRQDA/

[5] https://github.com/haichao592/UnAnsQ/

[6] https://github.com/dayihengliu/CRQDA/blob/master/pytorch-transformers-master/examples/run_fine_tune_bert_with_crqda.sh

| Training Data | EM ($\uparrow$) | F1 ($\uparrow$) |
|---|---|---|
| Baseline (no aug.) | $78.0_{\pm 0.3}$ | $81.2_{\pm 0.4}$ |
| + Antonym (no filtering) | $79.1_{\pm 0.3}$ | $82.1_{\pm 0.3}$ |
| + Antonym (random) | $79.0_{\pm 0.2}$ | $82.1_{\pm 0.2}$ |
| + Antonym (ppl) | $\mathbf{79.3}_{\pm 0.2}$ | $\mathbf{82.4}_{\pm 0.3}$ |
| + Entity (no filtering) | $80.1_{\pm 0.4}$ | $83.1_{\pm 0.2}$ |
| + Entity (random) | $\mathbf{80.7}_{\pm 0.1}$ | $\mathbf{83.6}_{\pm 0.0}$ |

Table 7: Comparing different filtering strategies on their SQuAD 2.0 development set performance (EM/F1) when fine-tuning BERT$_{\text{large}}$. Results are averaged over 3 random seeds. Given multiple augmented candidates generated from one question, "random" means we randomly sample one candidate, and "ppl" means we select the candidate with the lowest GPT-2 perplexity.

For antonym augmentation, we try random sampling and perplexity-based sampling in addition to using all of the generated data. All strategies improve over the baseline, but random sampling is marginally better than no filtering, and perplexity-based sampling is the best strategy.

For entity augmentation, we only compare two strategies: random sampling and no filtering. We do not try perplexity-based sampling as entity changes seem to impact perplexity in non-intuitive ways. Again, both strategies improve over the baseline but random sampling is better than no filtering.

## B.2 Combining augmentation strategies

Table 8 shows the results of combining antonym and entity augmentation. We see statistically significant improvements on 4 out of 6 models. Although these are good results when seen in isolation, we found that they did not show much of an improvement over just using entity augmentation. This suggests that it is worth exploring ways to more effectively combine the two strategies.

## C Annotation instructions

Together with this annotation protocol, you have received a link to a spreadsheet. The sheet contains 2 data columns and 3 task columns. The data columns consist of paragraphs and questions. In the paragraph column, each paragraph is prefaced with its topic. There are 100 paragraphs about various topics and 4 questions per paragraph, for a total of **400 data points**. You are asked to annotate the questions for the tasks of **unanswerability, relatedness, and readability**. Please be precise and consistent in your assignments. The columns have

built-in data validation and we will perform further tests to check for consistent annotation. Task-specific information is provided below.

## C.1 Unanswerability

For each of the 4 questions pertaining to a paragraph, please annotate **unanswerability based on the paragraph on a 0-1 scale** as follows:

- 0: answerable based on the paragraph

- 1: unanswerable based on the paragraph

Please note that you need to *rely exclusively on the paragraph* for this annotation, i.e., we are not interested in whether a question is answerable or unanswerable in general, but specifically whether the paragraph contains the answer to the question.

Ignore grammatical errors, changes in connotation, and awkward wording within questions if they do not obscure meaning.

Please pay attention to affixation (e.g., negation) that changes the meaning of a term.

When negation appears in a question, you should use the logical definition of negation, i.e., anything in the universe that isn't X counts as answering "What isn't X?" However, for this task, the universe is restricted to specific answers from the paragraph. As an example:

- Paragraph: *Cinnamon and Cumin are going out for lunch. Cinnamon will drive them there.*

- Question: *Who isn't Cinnamon?*
  => 0 (answerable) with "Cumin," who can be inferred to be another person mentioned in the paragraph who isn't Cinnamon

- Question: *Where isn't lunch?*
  => 1 (unanswerable), because there are no candidate answers in the paragraph that it would make sense to answer this question with

Some more examples:

- Paragraph: *Cinnamon and Cumin are going out for lunch. Cinnamon will drive them there.*

- Question: *Can Cinnamon drive?*
  => 0 (answerable)

- Question: *Can Cumin drive?*
  => 1 (unanswerable)

- Question: *cinammon can drive?*
  => 0 (answerable), despite the odd syntax and the typo

| Model | Baseline (no aug.) | | Combined | |
|---|---|---|---|---|
| | EM ($\uparrow$) | F1 ($\uparrow$) | EM ($\uparrow$) | F1 ($\uparrow$) |
| $BERT_{base}$ | $72.7_{\pm0.3}$ | $76.0_{\pm0.3}$ | $76.1_{\pm0.3}$ | $78.9_{\pm0.4}$ |
| $BERT_{large}$ | $78.0_{\pm0.3}$ | $81.2_{\pm0.4}$ | $80.6_{\pm0.3}$ | $83.5_{\pm0.3}$ |
| $RoBERTa_{base}$ | $78.7_{\pm0.1}$ | $81.8_{\pm0.1}$ | $79.9_{\pm0.1}$ | $82.8_{\pm0.1}$ |
| $RoBERTa_{large}$ | $85.8_{\pm0.2}$ | $88.8_{\pm0.2}$ | $85.7_{\pm0.1}$ | $88.5_{\pm0.1}$ |
| $ALBERT_{base}$ | $79.3_{\pm0.1}$ | $82.4_{\pm0.1}$ | $79.7_{\pm0.1}$ | $82.6_{\pm0.0}$ |
| $ALBERT_{large}$ | $82.1_{\pm0.2}$ | $85.2_{\pm0.1}$ | $82.2_{\pm0.3}$ | $85.0_{\pm0.2}$ |

Table 8: Combining our augmentation methods on SQuAD 2.0 shows significant improvements (coloured cells) across models according to a Welch's t-test ($\alpha$ = 0.05). Results (EM/F1) are averaged over 3 random seeds.

- Question: *lunch drive what?*
  => 1 (unanswerable), because the errors result in an incomprehensible sentence

## C.2 Relatedness

For each of the 4 questions pertaining to a paragraph, you need to annotate **relatedness to the paragraph on a 0-1** scale as follows:

- 0: unrelated to the paragraph

- 1: related to the paragraph

For a question to be related to a paragraph, **all parts of it should be related** to what the paragraph discusses. If any parts of the question are unrelated to the contents of the paragraph, please annotate it as unrelated.

If words in a question are in the paragraph even if they're combined in different ways that potentially don't make sense, this still counts as related. For instance, mixtures of names created using components of names in the paragraph count as related, but an entirely new made-up name would be unrelated.

Numbers and dates can be different from the ones mentioned in the question - this still counts as related.

Events that are related to the lives of people or history of companies (e.g., births, deaths, etc.) should be marked as related.

Ignore grammatical errors, changes in connotation, and awkward wording within questions if they do not obscure meaning. Some examples:

- Paragraph: *Cinnamon and Cumin are going out for lunch. Cinnamon will drive them there.*

- Question: *Can Cinnamon drive?*
  => 1 (related)

- Question: *Can Cumin drive?*
  => 1 (related)

- Question: *cinammon can drive?*
  => 1 (related), despite odd syntax and typo

- Question: *lunch drive what?*
  => 1 (related), because "lunch" and "drive" both appear in the paragraph despite the incomprehensibility of the question

- Question: *What sunscreen do bees use?*
  => 0 (unrelated), because the paragraph has nothing to do with sunscreen or bees

- Question: *When was Cumin born?*
  => 1 (related), because birth is related to a person's existence.

Some more examples of edge cases:

- Question: *What car does Cinnamon use?*
  => 1 (related), because Cinnamon is mentioned in the paragraph and cars are related to driving

- Question: *What food will Cinnamon and Cumin eat?*
  => 1 (related), because Cinnamon and Cumin are mentioned in the paragraph and food is related to their lunch plans

- Question: *What sunscreen do Cinnamon and Cumin use?*
  => 0 (unrelated), since sunscreen is unrelated to driving and eating

- Question: *Do bees drive to lunch?*
  => 0 (unrelated), since the paragraph does not discuss bees

- Question: *Do you want to go out to lunch?*
  => 0 (unrelated), because the paragraph is not about you

## C.3 Readability

For each question, you need to annotate **readability and fluency on a 1-3 scale** as follows:

- 1: incomprehensible

- 2: minor errors that do not obscure the meaning of the question (such as typos, agreement errors, missing words or extra words)

- 3: fluent questions

Please **focus on how syntactically well-formed a question is** without worrying about the meaning making sense. For example, "Do clouds watch television?" is a syntactically fluent question even if it does not make sense semantically.

Please ignore extra spaces and capitalization errors when they do not change the meaning of the question. Some examples:

- Paragraph: *Cinnamon and Cumin are going out for lunch. Cinnamon will drive them there.*

- Question: *Can Cinnamon drive?*
  => 3 (fluent question)

- Question: *can cumin drive?*
  => 3 (fluent question), despite the lack of capitalization

- Question: *What sunscreen do bees use?*
  => 3 (fluent question)

- Question: *cinammon can drive?*
  => 2 (minor errors), because of the typo in the name

- Question: *Does bees drive?*
  => 2 (minor errors), because the question is comprehensible even though it has an agreement error between "does" and "bees"

- Question: *lunch drive what?*
  => 1 (incomprehensible)

- Question: *Can lunch drive?*
  => 3 (fluent question) syntactically, even though it is semantically nonsensical.

## D More examples of augmented data

We present more examples of data generated using our augmentation strategies in Figures 3 and 4, along with the context paragraph and the answerable seed questions from SQuAD 2.0.

**Context:** *In 1952, following a referendum, Baden, Württemberg-Baden, and Württemberg-Hohenzollern merged into Baden-Württemberg. In 1957, the Saar Protectorate rejoined the Federal Republic as the Saarland. German reunification in 1990, in which the German Democratic Republic (East Germany) ascended into the Federal Republic, resulted in the addition of the re-established eastern states of Brandenburg, Mecklenburg-West Pomerania (in German Mecklenburg-Vorpommern), Saxony (Sachsen), Saxony-Anhalt (Sachsen-Anhalt), and Thuringia (Thüringen), as well as the reunification of West and East Berlin into Berlin and its establishment as a full and equal state. A regional referendum in 1996 to merge Berlin with surrounding Brandenburg as "Berlin-Brandenburg" failed to reach the necessary majority vote in Brandenburg, while a majority of Berliners voted in favour of the merger.*

**Answerable seed question:**
*Why did a regional referendum in 1996 to merge Berlin with surrounding Brandenburg fail?*
**Entity swapping:**
*Why did a regional referendum in 1996 to merge Berlin with surrounding West Pomerania fail?*

**Answerable seed question:**
*In 1957, the Saar Protectorate rejoined the Federal Republic as which city?*
**Entity swapping:**
*In 1957, the Saar Protectorate rejoined the Hohenzollern as which city?*

Figure 3: Further examples of entity augmentation of answerable seed questions.

**Context:** *Long distance migrants are believed to disperse as young birds and form attachments to potential breeding sites and to favourite wintering sites. Once the site attachment is made they show high site-fidelity, visiting the same wintering sites year after year.*

**Answerable seed question:**
*When do long distance migrants disperse?*
**Antonym swapping:**
*When do short distance migrants disperse?*

**Answerable seed question:**
*What do young birds form attachments to?*
**Antonym swapping:**
*What do old birds form attachments to?*

Figure 4: Further examples of antonym augmentation of answerable seed questions.