# OpenReview forum: "A Lightweight Method to Generate Unanswerable Questions in English"
_EMNLP/2023/Conference — EMNLP 2023 Findings_

### Official Review · Reviewer_iyDW · 2023-08-03

**Soundness:** 3

**Excitement:**

2: Mediocre: This paper makes marginal contributions (vs non-contemporaneous work), so I would rather not see it in the conference.

**Paper Topic And Main Contributions:**

The authors presented a training-free, simple and effective method through antonym and entity swap to create unanswerable data in the QA dataset. In addition, the proposed method achieved higher readability and relatedness than existing methods and higher final performance.


**Questions For The Authors:**


1. Can the performance of the proposed method be maintained even for more complex multi-hop datasets like HotpotQA?

2. Even when applying the proposed method of antonym and entity swaps, it seems that there is a possibility of still having answerable questions beyond the situations mentioned in the limitations. Have you taken into consideration or been aware of such scenarios?






**Reasons To Accept:**

1. well-written paper, and easy-to read.
2. simple, but effective method.
3. various baseline models and experimental analysis with significant test.



**Reasons To Reject:**

1. Less excitement by utilizing a methodology commonly used in data augmentation.
2. Experiments on limited dataset (only one SQuAD 2.0 dataset). The authors can use other datasets like NewsQA and TydiQA which contain unanswerable questions.
3. Despite the lower unanswerability in human evaluation (Table 4), the proposed method achieves high performance on unanswerable questions, as shown in Figure 2.  Analysis of this phenomenon is lacking.


**Reproducibility:**

4: Could mostly reproduce the results, but there may be some variation because of sample variance or minor variations in their interpretation of the protocol or method.

**Reviewer Confidence:**

4: Quite sure. I tried to check the important points carefully. It's unlikely, though conceivable, that I missed something that should affect my ratings.

---

> ### Author Rebuttal · Authors · 2023-08-27
>
> We thank the reviewer for their time and effort and address their questions and concerns below. We especially appreciate their recognition of our writing, the simplicity of our methodology, and our rigorous testing across multiple models and runs, along with significance testing.
>
> ## “Only one evaluation dataset (SQuAD)”
>
> 1. To show that our method is better and more lightweight than prior methods in the literature, we had to use the same dataset they do, which is also only SQuAD.
> 2. To show the raw benefits of our data augmentation method compared to no augmentation, we have **new results on the English portion of TydiQA** that corresponds to minimum-span extractive QA. We used BERT-large and BERT-base fine-tuned over three runs and both antonym augmentation and entity augmentation had **statistically significant improvements** over the baselines, with F1 and EM improving by 8-10 points on average.
>
> | Methods          | EM        | F1        |
> |------------------|-----------|-----------|
> | BERT-large       | 51.4±0.8  | 54.4±0.7  |
> | + Entity (ours)  | 60.7±1.4* | 62.6±1.7* |
> | + Antonym (ours) | 61.2±0.3* | 63.7±0.4* |
> | BERT-base        | 48.5±0.5  | 51.6±0.7  |
> | + Entity (ours)  | 58.9±1.2* | 61.2±1.4* |
> | + Antonym (ours) | 58.7±0.7* | 61.4±0.7* |
>
> We thank the reviewer for suggesting TydiQA and hope that the addition of this result to our suite of significance tests across model types on SQuAD convinces all the reviewers of the soundness of our claims.
>
> ## “The methodology is less exciting”
>
> While we agree that swaps are a common simple data augmentation method and perhaps not very exciting, our contribution is less about the method itself and more about showing its effectiveness compared to the prior literature which proposes more “exciting” methodology but is ultimately over-engineered for the problem.
>
> Publishing our work would further the field by establishing a simple, strong baseline for unanswerable question generation in the literature, one that future work in this area would have to justify its complexity against. This is a possibility we are excited about.
>
> ## “Lower unanswerability of our generated questions vs. higher performance on unanswerable questions”
>
> It is indeed interesting and seemingly counterintuitive that we outperform CRQDA despite the unanswerability of our generated questions (78%) being lower than theirs (91%). We have two hypotheses for why this is the case:
> 1. As Table 4 shows, the human-judged unanswerability of **our method is comparable with human-written unanswerable questions** (78% vs. 83%), suggesting that the setup intrinsically has some level of tolerance to data noise.
> 2. **CRQDA questions do not provide a high-quality training signal** because their generated questions are mostly unintelligible (52% less readable than human-written questions, compared to our 7%) and unrelated to the paragraphs (36% less related to human-written questions compared to our equivalent-to-human-written relatedness). This suggests a **compositional effect**; unanswerability, relatedness and readability all play a role _together_ and doing reasonably well at all of them is important.
> Although we hint at this in L237-243, we regret that we were unable to address this in more detail in the draft. We thank the reviewer for raising this point and will certainly add this to the camera-ready version.
>
>
> -------------------------------------------------
>
>
> ## Question A: Can performance be maintained for complex multi-hop datasets like HotpotQA?
>
> Probably! Although we would have to verify this empirically (and we are excited to pursue future work around this), we hypothesize that yes, our method would work very well for an entity-heavy dataset like HotpotQA, generating highly related and readable unanswerable questions. As an example, take the following question and supporting paragraphs from the dataset:
>
> **Q**: *Risingson is the first single from what album by Massive Attack, that was the first to be produced by Neil Davidge, along with the group?*
>
> **Paragraph A: Risingson**
> *"Risingson" is a song by the British trip hop group Massive Attack, released as a single on 7 July 1997. It is the first single from their third album "Mezzanine" and the eighth single overall.*
>
> **Paragraph B: Mezzanine (album)**
> *Mezzanine is the third studio album by English trip hop group Massive Attack, released on 20 April 1998 by Circa and Virgin Records. It was the first album to be produced by Neil Davidge, along with the group. The entire album was provided on their website for legal download many months before the physical release was announced, one of the first major uses of the MP3 format by a commercial organisation.*
>
> Antonym-swapping would produce the highly related, readable and unanswerable question:
>
> - *Risingson is the first single from what album by Massive Attack, that was the **last** to be produced by Neil Davidge, along with the group?*
>
> Entity-swapping (according to the NER output on this example) would give us even more options, such as:
>
> - ***Circa** is the first single from what album by Massive Attack, that was the first to be produced by Neil Davidge, along with the group?*
>
> - *Risingson is the first single from what album by **MP3**, that was the first to be produced by Neil Davidge, along with the group?*
>
> ## Question B: “Are you aware of scenarios where answerable questions are generated beyond the situations mentioned in the limitations?”
>
> Yes, we are aware of a few more error types due to the heuristic of using swaps and explain some of them below. We are adding these to the Limitations section of the paper in a subsection about the limited diversity and lack of deep semantics of our method, on the suggestion of this reviewer and another.
> - **Conjunctions**: With a context such as, “Devin and Anneke like milk chocolate.” and a seed question like “What kind of chocolate does Devin like?”, entity augmentation would generate a question like “What kind of chocolate does Anneke like?” which is still answerable.
> - **Commutative relations**: With a context such as “Beyoncé married Jay-Z in 2008.” and a seed question like “When did Beyoncé get married?”, entity augmentation would generate a question like “When did Jay-Z get married?” which is still answerable because marriage is commutative between two people.
> - **Information is elsewhere in the paragraph**: With a context such as “Twilight Princess was launched in North America in November 2006, and in Japan, Europe, and Australia the following month.” and a seed question like “When was Twilight Princess launched in North America?”, entity augmentation would generate a question like “When was Twilight Princess launched in Japan?” because the information is contained in the rest of the sentence.
> - **Other forms of yes/no questions**: Although we do not antonym-augment the most common forms of yes/no questions, we currently do not filter out questions like, “Outdoor cats are a bigger threat to birds than habitat loss—true or false?” and there are some questions of this type in the SQuAD dataset.

---

### Official Review · Reviewer_YiU3 · 2023-08-03

**Soundness:** 4

**Excitement:**

3: Ambivalent: It has merits (e.g., it reports state-of-the-art results, the idea is nice), but there are key weaknesses (e.g., it describes incremental work), and it can significantly benefit from another round of revision. However, I won't object to accepting it if my co-reviewers champion it.

**Paper Topic And Main Contributions:**

A simple yet effective approach is introduced to generate unanswerable questions for extractive question answering systems. Drawing inspiration from human-written unanswerable questions, the method involves performing antonym and entity swaps on answerable questions, thus producing unanswerable variants. The proposed technique surpasses the performance of baseline methods and achieves notable results in human-judged unanswerability, relatedness, and readability metrics. The paper highlights the practical usefulness of the method for building better question answering systems and establishes it as a strong baseline for future research.

**Questions For The Authors:**

- Question A: In Figure 2, as the number of augmented unanswerable questions increases, it creates a tradeoff that impacts the model's performance on answerable questions. Apart from overfitting on unanswerable questions, could you elaborate on the underlying reasons for this effect? Are there distinct characteristics or linguistic patterns present in the generated unanswerable questions that could potentially interfere with the model's accuracy in answering answerable questions?

**Reasons To Accept:**

- Simple, Efficient, and Effective: The method is lightweight and training-free. It is based on simple antonym and entity swaps on answerable questions, reducing the need for costly manual annotation or complex paraphrasing techniques. The strategy outperforms baseline methods and achieves notable results in human-judged unanswerability, relatedness, and readability metrics
- Potential Generalizability: The approach shows potential for easy extension to datasets beyond SQuAD 2.0, indicating its potential applicability to other question-answering tasks.

**Reasons To Reject:**

- Limited Diversity and Lack of Semantic Understanding: The proposed method relies on surface-level manipulations of the input questions (antonym and entity swaps) rather than having a deep understanding of the semantics and context of the questions. While this might yield some unanswerable questions, it could limit the diversity of the generated questions. There could be other linguistic variations and structures that are characteristic of unanswerable questions, which the method may not explore effectively.
- Overfitting to Training Data: The automatic method is based on SQuAD 2.0, which is a specific dataset that might have its own biases and characteristics. Depending solely on this dataset for training may lead to overfitting, and the generated questions might not generalize well to other domains or datasets.

**Reproducibility:**

4: Could mostly reproduce the results, but there may be some variation because of sample variance or minor variations in their interpretation of the protocol or method.

**Reviewer Confidence:**

4: Quite sure. I tried to check the important points carefully. It's unlikely, though conceivable, that I missed something that should affect my ratings.

---

> ### Author Rebuttal · Authors · 2023-08-27
>
> We thank the reviewer for their time and their acknowledgement of our method’s lightweightness, effectiveness across multiple metrics, and generalizability to datasets. Below, we address both issues they raised and answer their question. We hope they find our responses and new results compelling enough to raise their scores, or we request them to clarify which “minor points may need extra support or details.”
>
> ## “Overfitting to Training Data”
>
> First, we would like to remind the reviewer that as our augmentation method is training-free, **neither training nor overfitting is technically possible** - it can be applied to any dataset.
>
> 1. To show that our method is better and more lightweight than prior methods in the literature, we had to _evaluate_ on the same dataset they do, which is also only SQuAD. We discuss the implications of this in our Limitations section in the hope that this provides practical guidance for those trying to use our method on QA domains with different characteristics.
> 2. To show the raw benefits of our data augmentation method compared to no augmentation, we have **new results on the English portion of TydiQA** that corresponds to minimum-span extractive QA. We used BERT-large and BERT-base fine-tuned over three runs and both antonym augmentation and entity augmentation had **statistically significant improvements** over the baselines, with F1 and EM improving by 8-10 points on average.
>
> | Methods          | EM        | F1        |
> |------------------|-----------|-----------|
> | BERT-large       | 51.4±0.8  | 54.4±0.7  |
> | + Entity (ours)  | 60.7±1.4* | 62.6±1.7* |
> | + Antonym (ours) | 61.2±0.3* | 63.7±0.4* |
> | BERT-base        | 48.5±0.5  | 51.6±0.7  |
> | + Entity (ours)  | 58.9±1.2* | 61.2±1.4* |
> | + Antonym (ours) | 58.7±0.7* | 61.4±0.7* |
>
> We thank reviewer iyDW for suggesting TydiQA and hope that the addition of this result to our suite of significance tests across model types on SQuAD convinces all reviewers of the soundness of our claims.
>
> ## “Limited diversity and lack of semantic understanding”
>
> We thank the reviewer for bringing up the limited diversity of our generated examples. We are adding this to our Limitations section as it means that there will always be a qualitative gap between our samples and human-written samples, but want to emphasize that **this does not weaken our claims about outperforming previous automatic systems on SQuAD performance, readability, unanswerability and relevance to the topic**, which are arguably more important in this context.
>
> It is also true that our surface-level method (using heuristics) lacks semantic understanding, but so do the other automatic methods in this area. We will nevertheless add this point to our Limitations section as well, along with the following examples of failure modes of surface-level heuristics, on the suggestion of this reviewer and another.
> - **Conjunctions**: With a context such as, “Devin and Anneke like milk chocolate.” and a seed question like “What kind of chocolate does Devin like?”, entity augmentation would generate a question like “What kind of chocolate does Anneke like?” which is still answerable.
> - **Commutative relations**: With a context such as “Beyoncé married Jay-Z in 2008.” and a seed question like “When did Beyoncé get married?”, entity augmentation would generate a question like “When did Jay-Z get married?” which is still answerable because marriage is commutative between two people.
> - **Information is elsewhere in the paragraph**: With a context such as “Twilight Princess was launched in North America in November 2006, and in Japan, Europe, and Australia the following month.” and a seed question like “When was Twilight Princess launched in North America?”, entity augmentation would generate a question like “When was Twilight Princess launched in Japan?” because the information is contained in the rest of the sentence.
> - **Other forms of yes/no questions**: Although we do not antonym-augment the most common forms of yes/no questions, we currently do not filter out questions like, “Outdoor cats are a bigger threat to birds than habitat loss—true or false?” and there are some questions of this type in the SQuAD dataset.
>
> -----------------------------------------------
>
> ## Question A: What are the underlying reasons for the tradeoff between unanswerable and answerable questions?
>
> - As the reviewer points out in their question, we believe overfitting to be one of the main reasons for this effect.
> - Another potential reason for the tradeoff is noise, i.e., generated questions that are labelled unanswerable but are actually answerable. This could lead the model to abstain more on answerable questions, especially since our swapping means that both types of questions are often syntactically very similar. In column 1 of table 4, we quantify this noise as being about 20%.
> - These are hypotheses but we are interested in investigating them empirically in future work.

---

### Official Review · Reviewer_HhfT · 2023-08-10

**Soundness:** 3

**Excitement:**

3: Ambivalent: It has merits (e.g., it reports state-of-the-art results, the idea is nice), but there are key weaknesses (e.g., it describes incremental work), and it can significantly benefit from another round of revision. However, I won't object to accepting it if my co-reviewers champion it.

**Paper Topic And Main Contributions:**

This study publishes a data-enhanced approach to generate unanswerable questions to increase the performance of reading comprehension models. And this method has proven to promote the performance of the reading comprehension model when evaluated on the development set of the SQuAD 2.0 dataset.

**Reasons To Accept:**

This study published a simple and lightweight method of generating unanswered questions for experimental implementation and improved performance.

**Reasons To Reject:**

The article has a few weak points:
- In section 3.1, the author talked about finding and replacing antonyms, but he did not describe the method of identifying words and finding antonyms?
- At experiment and show the results, it would be better if the authors prove this method improves performance on more datasets.
- I don't really see the strength of this method when applied to practical applications.
- The contribution of the problem is really very simple and small, but has not brought a great value.

**Reproducibility:**

4: Could mostly reproduce the results, but there may be some variation because of sample variance or minor variations in their interpretation of the protocol or method.

**Reviewer Confidence:**

4: Quite sure. I tried to check the important points carefully. It's unlikely, though conceivable, that I missed something that should affect my ratings.

---

> ### Author Rebuttal · Authors · 2023-08-27
>
> We thank the reviewer for their time in reviewing our paper and address their comments below.
>
> ## “The contribution of the problem is really very simple and small, but has not brought a great value.”
>
> Respectfully, we disagree with this characterization of our work and see plenty of value in our proposed method, which:
> 1. Beats the previous state-of-the-art approaches to unanswerable question generation on SQuAD dev set performance
> 2. Beats prior methods on generating highly readable and highly related unanswerable questions
> 3. Is more data-efficient than prior methods as shown in our ablation study and analysis in Figure 2
> 4. Is completely training-free (0 parameters) and transferable compared to prior state-of-the-art method (593M parameters)
>
> We would appreciate it if the reviewer could clarify precisely what more they would like to see from our contribution.
>
> ## “I don’t really see the strength of this method when applied to practical applications.”
>
> - As motivated in our introduction in L31-39, the task of unanswerable question generation is applied practically to build more reliable QA systems.
> - Unanswerable questions do not exist in most QA datasets and are expensive and slow for humans to write, but we expect QA systems built on such datasets to be able to identify them.
> - Our data augmentation method helps to cheaply build more reliable QA systems, and improves performance as mentioned above.
>
> ## “Only one evaluation dataset (SQuAD)”
>
> 1. To show that our method is better and more lightweight than prior methods in the literature, we had to use the same dataset they do, which is also only SQuAD.
> 2. To show the raw benefits of our data augmentation method compared to no augmentation, we have **new results on the English portion of TydiQA** that corresponds to minimum-span extractive QA. We used BERT-large and BERT-base fine-tuned over three runs and both antonym augmentation and entity augmentation had **statistically significant improvements** over the baselines, with F1 and EM improving by 8-10 points on average.
>
> | Methods          | EM        | F1        |
> |------------------|-----------|-----------|
> | BERT-large       | 51.4±0.8  | 54.4±0.7  |
> | + Entity (ours)  | 60.7±1.4* | 62.6±1.7* |
> | + Antonym (ours) | 61.2±0.3* | 63.7±0.4* |
> | BERT-base        | 48.5±0.5  | 51.6±0.7  |
> | + Entity (ours)  | 58.9±1.2* | 61.2±1.4* |
> | + Antonym (ours) | 58.7±0.7* | 61.4±0.7* |
>
> We thank reviewer iyDW for suggesting TydiQA and hope that the addition of this result to our suite of significance tests across model types on SQuAD convinces all reviewers of the soundness of our claims.
>
> ## “What is the method for identifying words and finding antonyms?”
>
> The types of words to augment (noun, adjective or verb) are identified using part-of-speech tagging with spaCy (L107). Antonyms and lemmas are accessed using WordNet through NLTK (L111). This point struck us more as a question than a reason for rejection from the reviewer.

---

### Official Review · Reviewer_1Std · 2023-08-11

**Soundness:** 2

**Excitement:**

3: Ambivalent: It has merits (e.g., it reports state-of-the-art results, the idea is nice), but there are key weaknesses (e.g., it describes incremental work), and it can significantly benefit from another round of revision. However, I won't object to accepting it if my co-reviewers champion it.

**Paper Topic And Main Contributions:**

The paper discusses the approach of augmenting the question with entity and antonym swap with respect to the context to make it unanswerable. The author proposed the training-free, lightweighted approach to improve the performance of encoder-only models like BERT to identify unanswerable questions.


**Reasons To Accept:**

The approach is simple to understand and shown to work well in identifying the unanswerable questions. The author provides a method that does not require training. The paper compared with two baselines and show that their method outperforms baselines in SQUAD 2.0.

**Reasons To Reject:**

- The novelty of the paper is limited. The idea of entity and antonym was introduced previously in UNANSQ paper.
- According to Table 1, different methods have different augmentations resulting in different number of generated samples. The comparison presentaed in Table 2 is not with common denominator, making it hard to directly compare the numbers.
- As authors mentioned in Limitations, the QA system considered in the paper is limited to SQAUD 2.0 dataset. Extending the experiments to consider retrieval augmented extractive QA system would show true potential of the method.

**Reproducibility:**

4: Could mostly reproduce the results, but there may be some variation because of sample variance or minor variations in their interpretation of the protocol or method.

**Reviewer Confidence:**

4: Quite sure. I tried to check the important points carefully. It's unlikely, though conceivable, that I missed something that should affect my ratings.

---

> ### Author Rebuttal · Authors · 2023-08-27
>
> We are grateful to the reviewer for their time and gratified that of the 3 issues they flagged in their review as reasons to reject, 2 are merely misunderstandings which we address below and 1 is a request for additional experiments which we have also performed. We hope the reviewer will reconsider their scores in light of our points below.
>
> ## “Limited novelty” / “antonyms and entities were introduced in UNANSQ”
>
> The UNANSQ paper (https://aclanthology.org/P19-1415/) **does not** propose antonym or entity augmentation. Their method is based on aligning human-written answerable and unanswerable questions from the original dataset and training a model on these pairs. Perhaps the reviewer misinterpreted Table 4 from their paper, which is in fact a human analysis of 100 system outputs to see what types of unanswerable questions were being generated by the system (mostly negated versions of the seed questions).
>
> ## “It is hard to directly compare the numbers in Table 2 due to different data sizes”
>
> This is _exactly_ the reason for **our ablation study in Figure 2**, controlling the amounts of data we were using from each method. Our empirical finding (as described in L220) is that our methods perform comparably with or better than more complex methods, even with lower, balanced amounts of data, like 500 samples or 1,000 samples. We will improve the signposting of this ablation study earlier on the paper.
>
> ## “Only one evaluation dataset (SQuAD)”
>
> 1. To show that our method is better and more lightweight than prior methods in the literature, we had to use the same dataset they do, which is also only SQuAD.
> 2. To show the raw benefits of our data augmentation method compared to no augmentation, we have **new results on the English portion of TydiQA** that corresponds to minimum-span extractive QA. We used BERT-large and BERT-base fine-tuned over three runs and both antonym augmentation and entity augmentation had **statistically significant improvements** over the baselines, with F1 and EM improving by 8-10 points on average.
>
> | Methods          | EM        | F1        |
> |------------------|-----------|-----------|
> | BERT-large       | 51.4±0.8  | 54.4±0.7  |
> | + Entity (ours)  | 60.7±1.4* | 62.6±1.7* |
> | + Antonym (ours) | 61.2±0.3* | 63.7±0.4* |
> | BERT-base        | 48.5±0.5  | 51.6±0.7  |
> | + Entity (ours)  | 58.9±1.2* | 61.2±1.4* |
> | + Antonym (ours) | 58.7±0.7* | 61.4±0.7* |
>
> We thank reviewer iyDW for suggesting TydiQA and hope that the addition of this result to our suite of significance tests across model types on SQuAD convinces all reviewers of the soundness of our claims.

---

### Meta-Review · Area_Chair_oHPq · 2023-09-20

**Recommendation:** 3

**Metareview:**

The paper proposes augmenting training data for extractive QA by incorporating unanswerable questions, created by swapping antonyms and entities found in answerable questions. Experiments, including ablation studies, demonstrate that the proposed approach enhances performance on SQuAD (as discussed in the paper) and TidyQA (as mentioned in the rebuttal). The paper is well-structured and easy to follow. Its findings, including experimental setting, can be of interest to the QA community.

---

### Decision · Program_Chairs · 2023-10-07

**Decision:**

Accept-Findings

**Comment:**

The paper proposes augmenting training data for extractive QA by incorporating unanswerable questions, created by swapping antonyms and entities found in answerable questions. Experiments, including ablation studies, demonstrate that the proposed approach enhances performance on SQuAD (as discussed in the paper) and TidyQA (as mentioned in the rebuttal). The paper is well-structured and easy to follow. Its findings, including experimental setting, can be of interest to the QA community.